# Detection and Characterization of Zoonotic Pathogens in Game Meat Hunted in Northwestern Italy

**DOI:** 10.3390/ani14040562

**Published:** 2024-02-07

**Authors:** Irene Floris, Andrea Vannuccini, Carmela Ligotti, Noemi Musolino, Angelo Romano, Annalisa Viani, Daniela Manila Bianchi, Serena Robetto, Lucia Decastelli

**Affiliations:** 1SC Sicurezza e Qualità degli Alimenti, Istituto Zooprofilattico Sperimentale del Piemonte, Liguria e Valle d’Aosta (IZSPLV), Via Bologna 148, 10154 Turin, Italy; irene.floris@izsto.it (I.F.); andrea.vannuccini@izsto.it (A.V.); noemi.musolino@izsto.it (N.M.); angelo.romano@izsto.it (A.R.); manila.bianchi@izsto.it (D.M.B.); lucia.decastelli@izsto.it (L.D.); 2Posto di Controllo Frontaliero (PCF), Ponte Caracciolo MONTITAL, 16126 Genova, Italy; 3National Reference Centre for Wild Animals Diseases (CeRMAS), SC Valle d’Aosta, Istituto Zooprofilattico Sperimentale del Piemonte, Liguria e Valle d’Aosta (IZSPLV), 11020 Aosta, Italy; annalisa.viani@izsto.it (A.V.); serena.robetto@izsto.it (S.R.); 4SS Patologie della Fauna Selvatica, Istituto Zooprofilattico Sperimentale del Piemonte, Liguria e Valle d’Aosta (IZSPLV), 11020 Aosta, Italy

**Keywords:** wildlife, zoonotic pathogens, public health, game meat, virulence gene, *Salmonella* spp., *Listeria monocytogenes*, *Yersinia enterocolitica*

## Abstract

**Simple Summary:**

Wildlife can host zoonotic pathogens and transfer them to humans via food of animal origin. Moreover, European Union legislation regulates the hygiene of food of animal origin. In the present study, we investigated major zoonotic bacterial and viral pathogens (*Salmonella* spp., *Yersinia enterocolitica*, *Listeria monocytogenes*, Shiga-toxin-producing *Escherichia coli* (STEC), and hepatitis E virus (HEV)) in samples of wildlife and game meat from northwest Italy. Only a few samples were found to be contaminated with zoonotic bacteria, but they carried pathogenicity and antibiotic-resistance genes. HEV was not detected in any of the samples. Notwithstanding the low frequency of zoonotic pathogens, hygiene rules during the manipulation of game meat are essential to ensure consumer food safety.

**Abstract:**

Wildlife can represent a reservoir of zoonotic pathogens and a public health problem. In the present study, we investigated the spread of zoonotic pathogens (*Salmonella* spp., *Yersinia enterocolitica*, *Listeria monocytogenes*, Shiga-toxin-producing *Escherichia coli* (STEC), and hepatitis E virus (HEV)) considering the presence of virulence and antibiotic resistance genes in game meat from animals hunted in northwest Italy. During two hunting seasons (2020 to 2022), samples of liver and/or muscle tissue were collected from chamois (n = 48), roe deer (n = 26), deer (n = 39), and wild boar (n = 35). Conventional microbiology and biomolecular methods were used for the detection, isolation, and characterization of the investigated pathogens. Two *L. monocytogenes* serotype IIa strains were isolated from wild boar liver; both presented fosfomycin resistance gene and a total of 22 virulence genes were detected and specified in the text. Eight *Y. enterocolitica* biotype 1A strains were isolated from chamois (2), wild boar (5), and deer (1) liver samples; all showed streptogramin and beta-lactam resistance genes; the virulence genes found were *myfA* (8/8 strains), *ymoA* (8/8), *invA* (8/8), *ystB* (8/8), and *ail* (4/8). Our data underscore the potential role of wildlife as a carrier of zoonotic and antibiotic-resistant pathogens in northwest Italy and a food safety risk for game meat consumers.

## 1. Introduction

Wildlife can transmit zoonotic infectious agents to humans; as such, it constitutes a major public health problem in habitats shared by wild animals, domestic animals, and humans, where it can facilitate the spread of diseases [1]. From the “One Health” perspective, wild animals are a potential reservoir of zoonotic pathogens: about 75% of human pathogens are zoonotic, 70% of which are linked to wild animals [2]. Knowledge of the state of the health of wildlife is, therefore, essential to minimize the effect on human health from the consumption of hunted game and to study the spread and transmission of pathogens that may occur through the ingestion of infected and/or contaminated meat. Moreover, consumption of game meat has increased in Europe and worldwide, mainly owing to the increased availability of wild animal species in many areas [3].

Regulation (EC) No 853/2004 of the European Parliament [4] lays down specific rules on the hygiene of food of animal origin. Annex III, section IV, specifies the training of hunters in health and hygiene (Chapter I) and the handling of large wild game (Chapter II) [4]. However, the regulation does not apply to primary production for private domestic use and to “hunters supplying small quantities of wild game or wild game meat directly to the final consumer or to laboratories attached to local retail or retail outlets supplying the consumer final” (Art 1, paragraph 3, letter e, Regulation (EC) 853/2004). Furthermore, Member States can establish national regulation of these categories. Differently, federal agencies in the United States apply no type of inspection of game meat, thus increasing the risk of foodborne disease, and there are no food safety standards [1].

Management of hunted game meat relies on the risk analysis of pathogens and the collection of data on prevalence and levels of contamination. Meat contamination occurs mainly during evisceration with the transfer of digestive tract bacteria for *Campylobacter* spp., *Clostridium perfringens*, *Salmonella enterica*, Shiga toxin-producing *Escherichia coli* (STEC), *Yersinia enterocolitica*, or through the skin for *Staphylococcus aureus* and *Listeria monocytogenes* [3,5].

*Salmonella* spp. is a Gram-negative, flagellated, facultative anaerobic bacteria that belongs to the *Enterobacteriaceae* family, and more than 2500 serotypes are known [6,7]. Furthermore, salmonellosis was the second most often reported zoonosis in 2021, according to the European Union One Health 2021 Zoonoses Report [8]. *Y. enterocolitica* is another member of the *Enterobacteriaceae* family and can be found in animal reservoirs, especially in swine. This pathogen can cause zoonotic disease (e.g., acute enteritis with self-limiting diarrhea, fatal necrotizing enterocolitis, and septicemia) [9]. *L. monocytogenes* is a Gram-positive, facultatively anaerobic bacteria and the causative agent of foodborne disease; three main serovars are known (1/2a, 1/2b, 4b) [10].

*Escherichia coli* is a Gram-negative bacteria found naturally in the environment, food, and gut of humans and animals [11]. Shiga-toxin-producing *E. coli* (STEC) can cause serious diseases in humans (e.g., bloody diarrhea, haemolytic-uremic syndrome (HUS)) [12]. *E. coli* O157:H7 is the most frequently involved serotype in human diseases, but more than 400 O:H types of STEC (mainly O26, O45, O103, O104, O111, O121, and O145) have been associated with infection [13]. Transmission of STEC to humans can occur through the ingestion of contaminated meat or water, contact with animal feces, or person-to-person contact [8]. It has been observed that *E. coli* infections in animals can be asymptomatic; therefore, wildlife may be a natural reservoir of these pathogens, leading to an increased risk of zoonosis [14].

Hepatitis E virus (HEV), an emerging foodborne zoonosis, is a positive-sense single-strand RNA virus of the *Hepeviridae* family [15,16]. Major reservoirs of HEV are swine, wild boar, and wild ruminants, especially deer [17]. There are 8 known genotypes of HEV [18]; however, only genotypes 1 and 2 have been found in humans, while genotypes 3 and 4 have been detected in animals (e.g., pigs and wild boar, deer, chamois, roe deer) [15,16,19]. Transmission of HEV occurs mainly via direct contact with infected animals and through the consumption of raw or contaminated food products, including wild boar meat. The liver of infected animals is the main source of infection, though viral DNA can also be detected in muscle tissue during the viremia phase, thus exposing consumers to the risk of zoonosis [20].

In the present study, we wanted to investigate zoonotic pathogens (*Salmonella* spp., *Y. enterocolitica*, *L. monocytogenes*, Shiga-toxin-producing *E. coli* (STEC), hepatitis E virus (HEV)) in wildlife and game meat from animals hunted in the Aosta Valley (northwest Italy).

Conventional microbiology methods and biomolecular methods were used for the detection and isolation of the main pathogens investigated. Furthermore, we employed high-throughput biomolecular methods (whole genome sequencing (WGS) and polymerase chain reaction (PCR)) for the genetic characterization of pathogens isolated from wild animals, including antibiotic resistance and virulence genes.

## 2. Materials and Methods

### 2.1. Sampling

For this study, we collaborated with the National Reference Centre for Wild Animal Diseases (CeRMAS) of Piedmont, Liguria, and the Aosta Valley Experimental Zooprophylactic Institute (IZSPLV). Tissue samples (liver and/or muscle) were taken from 148 wild ungulates collected during two hunting seasons (September 2020 to December 2022) and submitted to the Food Safety and Quality laboratories of IZSPLV. In particular, the following animals were selected: chamois (n = 48, *Rupicapra rupicapra*), deer (n = 39, *Cervus elaphus*), wild boar (n = 35, *Sus scrofa*), and roe deer (n = 26, *Capreolus capreolus*). Both liver and muscle samples could not be sampled in all animals due to the low availability provided by hunters. The total number of matrices sampled and analyzed is summarized in Section 3.

The study area was the Aosta Valley, located in the northwest range of the Italian Alps and inhabited by domestic ruminants and abundant wild ungulates. The hunted wild ungulates were delivered on the same day of shooting to the district centers of the reference territorial area for morphobiometric assessment. Afterward, a total of 100 g of liver and muscle tissue (diaphragm) was sampled at the game processing center, placed in sterile specimen cups, refrigerated at 4 °C, and delivered to the laboratory for analysis.

Lab analyses were carried out using accredited biomolecular methods (PCR) and/or conventional microbiology tests for *L. monocytogenes*, *Salmonella* spp., STEC, *Y. enterocolitica*, and HEV. Liver and muscle tissues were selected as matrices for food safety interest because they represent animal products.

### 2.2. Bacterial Isolation and Characterization

All microbiological methods used in this study were validated and accredited according to ISO 17025 [21]. The analysis performed for each sample depended on the quantitative availability of the matrix at the time of analysis. In particular, a total of 110 liver samples were tested for the detection of *L. monocytogenes*, 110 liver samples and 136 muscle samples for *Salmonella* spp., 101 liver samples for *Y. enterocolitica,* and 100 liver samples for STEC. The distribution of samples by animal species and matrices is described in Section 3.

Detection of *L. monocytogenes* in liver tissue samples was carried out according to ISO 11290-1:2017 [22] and AFNOR BRD 07/10-04/05. Samples were first screened using biomolecular testing; then, microbiological analyses were performed to identify the pathogen and to obtain isolated colonies. The analysis was performed on 25 g of liver, to which 225 mL Listeria Special Broth (LSB) (BioRad, Hercules, CA, USA) was added and incubated at 30 °C for 24 h. From the incubated broth, the iQ-Check Scan^®^ instrument (BioRad, Hercules, CA, USA), with the iQ-Check *Listeria monocytogenes* II^®^ kit (BioRad, Hercules, CA, USA), was used for DNA extraction and preparation of the amplification mix following the certified protocol AFNOR BRD 07/10-04/05. Subsequently, the amplification step was performed with the CFX96 Touch Real-Time PCR Detection System instrument (BioRad, Hercules, CA, USA).

Microbiological analyses were performed on the positive samples with the PCR method to confirm the presence of live and viable microorganisms in the matrices according to UNI EN ISO 11290-1:2017A. A primary enrichment step was performed on 25 g of sample in 225 mL of Demi-Fraser broth for 25 h at 30 °C; a secondary enrichment was performed with 100 µL of Demi-Fraser broth in 10 mL of Fraser broth for 24 h at 37 °C; both enrichments were then plated on Ottaviani Agosti Listeria Agar (ALOA) (Biolife, Monza, Milan, Italy) and Listeria PALCAM agar base (Biolife, Monza, Milan, Italy) media. Green/blue colonies surrounded by an opaque ring grown on ALOA medium and small (<1 mm) gray colonies surrounded by a black ring grown on PALCAM were considered positive.

Then, a multiplex PCR was performed on the isolates to identify serotype groups (IIa, IIb, IIc, Iva, IVb). For these purposes, *prfa* [23], *prs*, *lmo0737*, *lmo1118*, *orf 2819*, and *orf 2110* [24] genes were searched. The primers and amplification profiles were sourced from the bibliography (Table 1) [25]. Specifically, the following amplification protocol was followed: preheating at 95 °C for 15 min in one cycle; the amplification step was repeated for 35 cycles: denaturation at 95 °C for 30 s, annealing at 58 °C for 90 s, extension at 72 °C for 90 s; final extension step at 72 °C for 10 min in one cycle. Serotype determination is described in Appendix A.

Detection of *Salmonella* spp. in liver and muscle tissue samples was performed according to ISO 6579-1:2017 [26] and AFNOR BRD 07/6-07/04. The analysis was performed from 25 g of matrix to which 225 mL of BPW (Buffered Peptone Water) was added and incubated at 37 °C for 18 h. The next steps in the procedure for the detection of the pathogen *Salmonella* spp. are similar to those described for the detection of *L. monocytogenes*. In this case, the kit used for DNA extraction and master mix preparation was the iQ-Check *Salmonella* II^®^ kit (BioRad, Hercules, CA, USA) certified AFNOR BRD 07/6-07/04. The amplification step was performed with the CFX96 Touch Real-Time PCR Detection System instrument.

Microbiological analyses were performed on the positive samples with the PCR method according to UNI EN ISO 6579-1:2017 in order to confirm the presence of live and viable microorganisms in the matrices. Therefore, 100 µL of BPW pre-enrichment was added to 10 mL of Rappaport Vassiliadis Soy (RVS) (Oxoid, Milan, Italy) and incubated at 41.5 °C for 24 h; 1 mL of BPW pre-enrichment was added to 10 mL of Muller-Kauffmann tetrathionate-novobiocin (MKTTn) (Oxoid, Milan, Italy) and incubated at 37 °C for 24 h. At the end of the incubation, the enriched sample from both RVS and MKTTn was plated on two plates of Brilliant Green Agar (BGA) and Xylose CALysine Desoxycholate Agar (XLD) media each, both incubated at 37 °C for 24 h following the instructions of UNI EN ISO 6579:1/2017. Pink colonies grown on BGA medium and colonies with a black center and a transparent ring grown on XLD were considered positive.

Detection of Shiga toxin-producing *E. coli* (STEC) was carried out in liver tissue samples according to ISO/TS 13136:2012 [27]. The analysis was performed from 25 g of liver, to which 225 mL BPW was added and incubated at 41.5 °C for 24 h. The subsequent steps of the procedure for the detection of the pathogen STEC are similar to those described for the detection of *Salmonella* spp. and *L. monocytogenes*. The kit used for DNA extraction and master mix preparation was iQ-Check STEC VirX PCR Detection (BioRad, Hercules, CA, USA) AOAC certified No. 121203-2019. The amplification step of the pathogenicity genes (*stx1*, *stx2*, *eae*) was performed with the CFX96 Touch Real-Time PCR Detection System instrument.

Samples testing positive underwent analysis for virulence genes (*stx1*/*stx2*, *eae*) using a second Real-Time Multiplex PCR. The primers, probe (Table 2), and amplification profile were used in accordance with ISO/TS 13136:2012 [27]. Specifically, the following amplification protocol was followed: preheating at 95 °C for 15 min in one cycle; the amplification step was repeated for 40 cycles: denaturation at 95 °C for 10 s and annealing at 60 °C for 50 s. Positive samples from the latter analysis underwent PCR to identify the serogroup of the isolated strains (O157, O111, O26, O103, O145) [27].

Detection of *Y. enterocolitica* in liver samples was performed according to ISO 10273:2017 [28]. The analysis was performed from 25 g of matrix to which 225 mL of Buffered Saline Broth (PBS) (AMRESCO, VWR Int., Milan, Italy) was added and incubated at 25 °C for 44–48 h. Then, a Cefsulodin-Irgasan-Novobiocin (CIN) agar plate (Oxoid, Milan, Italy) was plated with the broth and incubated for 24–48 h at 30 ± 1 °C. After incubation, 10 mL of PSB Broth was added to 90 mL of Irgasan Ticarcillin Potassium Chlorate Broth (ITC) (Microbiol Diagnostici, Cagliari, Italy) and incubated at 25 °C for 44 h. Then, 500 µL of PSB and 500 µL of ITC were added to 4500 µL of potassium hydroxide (KOH) in two different test tubes, and from each, a plate of CIN was plated. Typical, small (<1 mm), smooth colonies with a red center and translucent edge were considered positive. Isolated strains were identified on a MALDI-TOF MS system (MALDI Biotyper Sirius, Bruker Daltonics GmbH & Co. KG, Bremen, Germany). Each strain was biotyped using miniaturized biochemical tests (API 20E and API 50 CH, Biomerieux, Marcy-l’Étoile, France) following the manufacturer’s instructions.

### 2.3. Hepatitis E Virus Detection

A total of 83 muscle samples were tested for the detection of HEV; the distribution of samples by animal species is described in Section 3. Detection of HEV was performed according to ISO 22174:2005 [29]. Virus concentration was performed following the protocol CCM 2016-HEV [30], “Protocol for the detection of Hepatitis E virus in food—Part1: Concentration of virus from food matrix”. The analysis was performed from 5 g of muscle sample that was subjected to homogenization and transferred to a sterile stomacher bag with a filter, to which 7 mL of Trizol was added. Then, the liquid part was recovered from the filter compartment and transferred into a Falcon tube that was centrifuged at 10,000× *g* for 20 min at 4 °C to settle the matrix residue. The supernatant was collected and transferred to another clean Falcon tube to which 1.4 mL of chloroform was added. The sample was first incubated for 15 min at room temperature and then centrifuged at 10,000× *g* for 15 min at 4 °C. In the end, the supernatant was recovered and stored at −20 °C until the nucleic acids were extracted.

DNA extraction was performed using an Invitrogen PureLink™ viral RNA Minikit (Thermo Fisher Scientific, Waltham, MA, USA) following the manufacturer’s instructions.

RNA retrotranscription and amplification were performed using an Invitrogen RNA Ultrasense One-step qRT-PCR system kit on a CFX96 Touch Real-Time PCR Detection System instrument using the primers sourced from the bibliography [31]: forward JVHEVF (GGT GGT TTC TGG GGT GAC) and reverse JVHEVR (AGG GGT TGG TTG GAT GAA) associated with the JVHEVP mod probe (FAM-TGA TTC TCA GCC CTT CGC-MGB) [31]. Specifically, the following protocol was followed: retrotranscription at 50 °C for 1 h; preheating at 95 °C for 5 min in one cycle; the amplification step was repeated for 45 cycles: denaturation at 95 °C for 15 s, annealing at 60 °C for 60 s, extension at 65 °C for 60 s.

### 2.4. DNA Extraction and Whole Genome Sequencing (WGS)

One strain per positive sample was isolated (8 strains of *Y. enterocolitica* and 2 strains of *L. monocytogenes*) for molecular analysis and stored at −20 °C. DNA extraction was performed using an Extractme Genomic DNA Isolation Kit (Blirt, Gdańsk, Poland) following the manufacturer’s instructions with modifications. The initial lysis step involved the treatment of lysozyme (10 mg/mL) as described below: 90 min at 37 °C for *L. monocytogenes* and 15 min at 37 °C for *Y. enterocolitica*. DNA quantity was measured with a Qubit fluorometer (Thermo Fisher Scientific). Library preparation was performed using an Illumina DNA Library Prep Kit (Illumina, San Diego, CA, USA), followed by sequencing with an Illumina MiSeq System Kit (Illumina) and MiSeq V3 chemistry in a run 2 × 151 bp paired-end reads, according to the manufacturer’s protocol.

### 2.5. Data Analysis of Whole Genome Sequencing (WGS)

Bioinformatic analysis was performed using the Galaxy interface [32,33]. The raw reads were trimmed using Trimmomatic 0.38 [34] for removing Nextera adaptors and other Illumina-specific sequences (Illuminaclip set to Nextera (paired-ended)), removing low-quality residues at the start and the end of the reads (leading:10 and trailing:10), clipping reads when the average Q-scores dropped below 20 over a sliding window of four residues (sliding window, 4:20), and dropping reads shorter than 40 bases after processing (minlen, 40). The trimmed reads were assembled de novo on a Unicycler 0.4.8.0 [35] for the bridging mode moderate contig size and misassembly rate (bridging mode set to Normal), and contigs less than 200 bp in length were excluded (exclude contigs from the FASTA file which are shorter than this length (bp) set to 200). The assembled genomes were then analyzed by the Center for Genomic Epidemiology (CGE) (accessed via https://www.genomicepidemiology.org/services/ accessed on 5 December 2023).

ResFinder (accessed via https://cge.food.dtu.dk/services/ResFinder/ accessed on 5 December 2023) was used to determine resistance genes in selected strains for both *Y. enterocolitica* and *L. monocytogenes*. The MLST 2.0 tool (accessed via https://cge.food.dtu.dk/services/MLST/ accessed on 5 December 2023) was used for identifying *Y. enterocolitica* strains sequence type (ST) [36]. VirulenceFinder 2.0 (accessed via https://cge.food.dtu.dk/services/VirulenceFinder/ accessed on 5 December 2023) was used to detect virulence genes in *L. monocytogenes* by setting the threshold for identification at 90% and the minimum length at 60% [37]. The virulence genes were compared with those listed in the Virulence Factors of Bacterial Pathogens (VFBP) database [37]. Detection of *Y. enterocolitica* virulence genes was performed using the Galaxy interface with the NCBI BLAST tool adherence (*myfA*), invasion (*ail*, *invA*), exotoxin (*ystA*, *ystB*), modulator genes (*ymoA*).

## 3. Results

### 3.1. Bacterial Isolation and Characterization

Results of molecular and microbiological analysis are summarized in Table 3 for liver samples and in Table 4 for muscle samples.

A total of 110 liver samples were analyzed for *L. monocytogenes*: 3 samples (2.7%) tested positive by molecular method; these were analyzed by classical microbiology methods for confirmation. Positivity was confirmed for 2 out of 3 samples (1.8%), from which two strains were isolated. Both strains were detected from wild boar samples.

The two positive strains were then tested by multiplex PCR for serotyping: both strains No. 1 and No. 2 tested positive for *prs*, *lmo0737,* and *lmo1118* genes. These, compared with the patterns in Appendix A, show that both strains are serotype IIa.

*Salmonella* spp. dectection was performed on 246 samples (*n* = 136 muscle and *n* = 110 liver). All 246 samples tested negative for *Salmonella* spp. by PRC method, and no other tests were performed.

Shiga toxin-producing *E. coli* (STEC) strain detection was performed on 100 liver samples. A total of 23 samples tested positive by the first molecular method applied. Then, confirmatory molecular testing for *stx1*, *stx2*, and *eae* genes was performed on one strain isolated from each sample. No STEC was detected, and no other tests were performed.

*Y. enterocolitica* detection was performed on 101 liver samples. A total of 8 strains (7.9%) were collected, all identified as biotype 1A. Specifically, 5 samples from wild boar, 2 from chamois, and 1 from deer tested positive for *Y. enterocolitica* 1A.

### 3.2. Hepatitis E Virus Detection

None of the 83 muscle tissue samples tested positive for HEV. Results are summarized in Table 4.

### 3.3. Molecular Characterization of Y. enterocolitica and L. monocytogenes

Table 5 presents the results of WGS. In all *Y. enterocolitica* strains, resistance genes for streptogramin and beta-lactam classes were found according to the ResFinder service provided by the CGE. Analyses performed on *Y. enterocolitica* strains, with the service provided by CGE cgMLST, showed that six assembled genomes belonged to strain type 1674, and two to strain type 1695. The *Y. enterocolitica* virulence genes found using the Galaxy interface with the NCBI BLAST tool were adherence (*myfA*), invasion (*ail*, *invA*), exotoxin (*ystB*), and modulator genes (*ymoA*). The analysis performed with the ResFinder service provided by the CGE showed that both *L. monocytogenes* strains were positive for resistance genes to fosfomycin. Analyses of *L. monocytogenes* strains performed by CGE MLST 2.0 showed that one assembled genome belonged to strain type 21 and one to strain type 451. Through the CGE VirulenceFinder tool, a total of 22 different virulence genes were found, which are summarized in Table 5. The genes found were then compared with those in the VFDB database, and the following functions have been associated: motility (*ActA*), adherence (*Ami*, *Lap*, *LapB*, *fbpA*, *inlF*), immune modulation (*OatA*), stress survival (*bsh*), invasion (*inlA*, *inlB*, *lpeA*, *vip*), and nutritional/metabolic factor (*IplA1*). The following genes were not found in the VFDB database: *lntA*, *pgdA*, *inlC*, *inlk*, *svpA*, *uHpt*, *prfA*, *prsA2*, *inlJ.*

## 4. Discussion

This study investigated the presence of pathogenic microorganisms (*Salmonella* spp., *L. monocytogenes*, *Y. enterocolitica*, STEC, HEV) in tissue samples from wild animals. Similar studies have already been conducted in the same geographical area in the past decade [3]. Despite the large number of samples analyzed, including both liver (110) and muscle (136), none tested positive for *Salmonella* spp.; this observation is shared by a study performed in Germany [38] but not by studies performed in central [39] and northern Italy [40]. Of note, however, is that the two Italian studies were only carried out on samples from wild boar, whereas this study included samples from other ungulates. Wild boar is a known reservoir of *Salmonella* spp. [41,42,43]. The animal’s adaptation to urban environments, direct contact with waste, and infected carcasses or farmed animals allows for continuous exposure to *Salmonella* spp., explaining its role as a reservoir for the pathogen [44].

The prevalence of *L. monocytogenes* was about 2%, in line with previous studies in the same geographical area [3], where wild boars were the most often infected species. Furthermore, both strains were serogroup IIa, as found by other European studies [45,46]. *L. monocytogenes* is known to be naturally resistant to fosfomycin [47], as shown in the results obtained. No other antibiotic resistance was found for the isolated strains, in contrast with findings reported by other studies [42]. For the characterization of *L. monocytogenes*, ST 21 and 451 were analyzed in respective isolates. Similar strains were reported in a study carried out in Finland [46]. The virulence factors found in the *L. monocytogenes* strains isolated in our study were heterogeneous, as found in other similar studies [46]. Some of them can be associated with genes regulating motility (*ActA*), adherence (*Ami*, *Lap*, *LapB*, *fbpA*, *inlF*), immune modulation (*OatA*), stress survival (*bsh*), invasion (*inlA*, *inlB*, *IpeA*, *vip*), and nutritional/metabolic factor (*IplA1*) [48]. The presence of these genes carries implications for the severity of infection that the pathogen can establish. For example, the presence of genes such as *ActA* (motility), *inlF* (adherence), and *inlA* (invasion) allows *L. monocytogenes* to penetrate inside the host cell and evade the immune system [48]. The *OatA* gene confers resistance to several antimicrobials, including lysozymes [49]. Since it can colonize the gastrointestinal tract, *L. monocytogenes* can survive acute bile toxicity, and genes such as *bsh* confer such resistance and allow the pathogen to persist in the gut [50]. The *IplA1* gene allows *L. monocytogenes* to benefit from host cell metabolism and proliferate in the cytoplasm [51].

Our data show a prevalence of *Y. enterocolitica* of about 10%, which is slightly higher than the prevalence of 2% reported for Liguria [52] but lower than that for Tuscany [42]. Beta-lactam antibiotic resistance genes were found in all strains of *Y. enterocolitica*; this type of resistance has also been confirmed phenotypically in the same area [52,53]. This suggests a circulation of strains potentially resistant to beta-lactam in northwest Italy. The presence of resistance genes for beta-lactams has also been found in other studies in different countries [54,55]. Moreover, as a Gram-negative microorganism, *Y. enterocolitica* shows an intrinsic resistance to streptogramin [56].

Molecular analysis results showed that many *Y. enterocolitica* strains were genetically related: six strains with cgST 1674 and two with cgST 1695. This may be explained by the small size of the geographical area where the samples were obtained.

The virulence gene *ail* has been observed sporadically in *Y. enterocolitica* biotype 1A [57,58], which is considered nonpathogenic [58,59]. The nonfunctional *Inv* gene has been observed in nonpathogenic strains of *Y. enterocolitica* 1A [60]. In the present study, WGS revealed the *inv* gene in all strains of *Y. enterocolitica* biotype 1A and the *ail* gene in 5 of the 8 strains. This observation is in line with data from a study conducted in Liguria in recent years [52]. Further studies are desirable to determine the functionality of such genes and their expression and the potential correlation between pathogenic *Y. enterocolitica* 1A strains and wild animals. Two liver samples from wild boar tested positive for both *L. monocytogenes* and *Y. enterocolitica*. Co-infection with two foodborne pathogens in the same game meat raises concern about consumer food safety.

None of the samples tested positive for STEC; this finding can be compared with a study conducted in Japan [61]. However, this pathogen has been detected in other countries [62,63].

No HEV was found in any of the samples, probably because the virus is detectable in muscle tissue only during the active viremia stage. While HEV testing is typically performed in the liver, we tested the muscle tissue because it is more suitable from a food safety perspective. Furthermore, previous studies conducted in Italy reported a prevalence of 1.35% and 5.6% in wild boar muscle [18] and muscle and liver [33], respectively.

Overall, the low frequency of *L. monocytogenes* and *Y. enterocolitica* and the absence of *Salmonella* spp., STEC, and HEV may be attributed to the sampling area where the animals are free-ranging species and rarely share pastures with domestic animals. Domestic livestock graze in mountainous areas only during the summer, thus limiting the potential transmission of pathogens from intensively farmed animals to wildlife.

Foodborne pathogens are frequently carried by animals or animal products and can be found in the entrails or intestine (*Salmonella* spp., STEC), soil, or surface water (*L. monocytogenes*) [64]. However, a lack of hygiene rules can increase the risk of food contamination.

To improve the hygiene of game meat, the entire carcass should be sent to a slaughterhouse or a game-handling facility regardless of the destination of the meat. Game-handling facilities can guarantee the correct handling of carcasses, and inspection is carried out by a competent agency. In the slaughterhouse, trained operators possess the necessary skills at each stage. In addition, an official veterinarian can ensure that the meat is safe for consumption [65].

## 5. Conclusions

Wildlife can represent a reservoir of zoonotic pathogens and a public health problem. The results obtained in this study show that, despite the low prevalence, there are pathogens such as *Y. enterocolitica* and *L. monocytogenes* circulating in the wildlife in northwestern Italy. Specifically, the liver is more at risk than the muscle, which tested negative for the pathogens investigated. Moreover, pathogens strains were isolated from wild boar, chamois, and deer, underlining the role of wildlife as a possible carrier of foodborne pathogens. The results achieved in our investigation showed that the pathogenic strains isolated from *Y. enterocolitica* and *L. monocytogenes* carry virulence genes and antibiotic resistance genes. This can lead to the establishment of more severe diseases, especially for hunters, wildlife stakeholders, and consumers. Moreover, data obtained by WGS methods did not show the presence of resistance genes to more than one class of antibiotics in any of the isolated strains, indicating the low level of circulation of resistance strains between wild animals.

Nevertheless, *Salmonella* spp., Shiga-toxin-producing *Escherichia coli,* and hepatitis E virus were not detected in any of the analyzed samples. This may be associated with the selected wildlife species and organ type sampled.

In conclusion, the application of control procedures and protocols is essential to ensure food safety for consumers and the health surveillance of wild animals. Meat safety should always be a top priority regardless of its intended use. Consequently, strict hygiene rules should be applied for the slaughter, manipulation, and production of game meat, sausages, or other cured meats made with the liver of wildlife species to ensure food safety.

## Figures and Tables

**Table 1 animals-14-00562-t001:** PCR primer sequences for the determination of *L. monocytogenes* serogroups.

Target	Primer Sequence (5′-3′)		Size
*prfa*	Forward	GAT ACA GAA ACA TCG GTT GGC	274
	Reverse	GTG TAA CTT GAT GCC ATC AGG	
*prs*	Forward	GCT GAA GAG ATT GCG AAA GAA G	370
	Reverse	CAA AGA AAC CTT GGA TTT GCG G	
*lmo0737*	Forward	AGG GCT TCA AGG ACT TAC CC	691
	Reverse	ACG ATT TCT GCT TGC CAT TC	
*lmo1118*	Forward	AGG GGT CTT AAA TCC TGG AA	906
	Reverse	CGG CTT GTT CGG CAT ACT TA	
*orf 2819*	Forward	AGC AAA ATG CCA AAA CTC GT	471
	Reverse	CAT CAC TAA AGC CTC CCA TTG	
*orf 2110*	Forward	AGT GGA CAA TTG ATT GGT GAA	597
	Reverse	CAT CCA TCC CTT ACT TTG GAC	

**Table 2 animals-14-00562-t002:** PCR primer and probe sequences for the detection of virulence genes of STEC.

Target	Oligo Sequence	
*stx1*	Primer Forward	TTT GTY ACT GTSA CAG CWG AAG CYT TAC G
	Primer Reverse	CCC CAG TTC ARW GTR AGR TCM ACR TC
	Probe	CTG GAT GAT CTC AGT GGG CGT TCT TAT GTA A
*stx2*	Primer Forward	TTT GTY ACT GTSA CAG CWG AAG CYT TAC G
	Primer Reverse	CCC CAG TTC ARW GTR AGR TCM ACR TC
	Probe	TCG TCA GGC ACT GTC TGA AAC TGC TCC
*eae*	Primer Forward	CAT TGA TCA GGA TTT TTC TGG TGA TA
	Primer Reverse	CTC ATG CGG AAA TAG CCG TTA
	Probe	ATA GTC TCG CCA GTA TTC GCC ACC AAT ACC

**Table 3 animals-14-00562-t003:** Overview of the number of liver samples tested by molecular and microbiological methods and number of positive samples divided by animal species and by microorganism.

		*Salmonella* spp.		*Listeria monocytogenes*		*Yersinia enterocolitica*		Shiga Toxin-Producing *E. coli* (STEC)	
Matrix	Wildlife	No. of Samples Tested	No. Positive	No. of Samples Tested	Positive	No. of Samples Tested	Positive	No. of Samples Tested	Positive
Liver	Chamois	35	0	34	0	31	2	34	0
	Roe deer	22	0	22	0	18	0	18	0
	Wild boar	27	0	28	2	27	5	24	0
	Deer	26	0	26	0	25	1	24	0
	Total	110	0	110	2	101	8	100	0

**Table 4 animals-14-00562-t004:** Overview of the number of muscle samples tested for *Salmonella* spp. and HEV with molecular methods and number of positive samples.

		*Salmonella* spp.		HEV	
Matrix	Wildlife	No. of Samples Tested	No. Positive	No. of Samples Tested	Positive
Muscle	Chamois	42	0	26	0
	Roe deer	21	0	15	0
	Wild boar	34	0	30	0
	Deer	39	0	12	0
	Total	136	0	83	0

**Table 5 animals-14-00562-t005:** Genetic identification, cgMLST profile (*Y. enterocolitica*), MLST profile (*L. monocytogenes*), genotypic antibiotic resistance, and virulence genes of analyzed *Y. enterocolitica* and *L. monocytogenes* isolated strains by Whole Genome Sequencing (WGS). CgMLST: Core Genome Multilocus Sequencing Typing; MLST: Multilocus Sequencing Typing; n.d.: not determined.

Strain	Species Identification	CgMLST	MLST	Genotypic Antibiotic Resistance	Virulence Genes
1	*Y. enterocolitica*	1674	n.d.	Streptogramin, beta-lactam	*ystB*, *myfA*, *ymoA*, *invA*
2	*Y enterocolitica*	1674	n.d.	Streptogramin, beta-lactam	*ystB*, *myfA*, *ymoA*, *invA*
3	*Y. enterocolitica*	1674	n.d.	Streptogramin, beta-lactam	*ail*, *ystB*, *myfA*, *ymoA*, *invA*
4	*Y. enterocolitica*	1695	n.d.	Streptogramin, beta-lactam	*ystB*, *myfA*, *ymoA*, *invA*
5	*Y. enterocolitica*	1695	n.d.	Streptogramin, beta-lactam	*ystB*, *myfA*, *ymoA*, *invA*
6	*Y. enterocolitica*	1674	n.d.	Streptogramin, beta-lactam	*ail*, *ystB*, *myfA*, *ymoA*, *invA*
7	*Y. enterocolitica*	1674	n.d.	Streptogramin, beta-lactam	*ail*, *ystB*, *myfA*, *ymoA*, *invA*
8	*Y. enterocolitica*	1674	n.d.	Streptogramin, beta-lactam	*ail*, *ystB*, *myfA*, *ymoA*, *invA*
9	*L. monocytogenes*	n.d.	21	Fosfomycin	*inlA*, *inlB*, *lpeA*, *lntA*, *pgdA*, *inlC*, *inlk*, *lplA1*, *svpA*, *uHpt*, *prfA*, *prsA2*
10	*L. monocytogenes*	n.d.	451	Fosfomycin	*ActA*, *Ami*, *Lap*, *LapB*, *OatA*, *bsh*, *fbpA*, *inlA*, *inlB*, *inlC*, *inlF*, *inlJ*, *inlk*, *lntA*, *lpeA*, *lplA1*, *pgdA*, *svpA*, *uHpt*, *vip*

## Data Availability

The genome datasets presented in this study can be found in the online repository Mendeley Data at the following link: https://doi.org/10.17632/vnht3t2znc.1 accessed on 8 December 2023.

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
