# Peer review of "Detection and Characterization of Zoonotic Pathogens in Game Meat Hunted in Northwestern Italy"

_animals, 2024, doi:10.3390/ani14040562_

Round 1

Reviewer 1 Report

Comments and Suggestions for Authors

This is an interesting article which investigates pathogen carriage within game meat from North West Italy. This is an under researched area and one of interest given the rapid rise in consumption of game meats. The paper itself is generally well written. I have a few grammatical suggests which I have detailed below. Also numbers seem to go all over the place in the results so perhaps needs a bit of justification in here.

Line 20- could almost just say North-West Italy in here and delete ‘area’

Line 22- perhaps don’t need ‘instead’ in here?

Line 33- both of which were resistance to Fosfomycin… (reword)

Line 70- facultative anaerobic bacterium of the …. (reword)

Line 71- space needed after reference

Line 72- salmonellosis doesn’t need capitalising

Line 218- this doesn’t make sense – perhaps ‘In detail, a total of 249 samples (N = 138 muscle and N = 111 liver) were obtained and analysed (table 5).’

Line 225- tested positive by the molecular …. (reword)

Line 230- is this a surprise as this is more commonly found in the gut?

Line 232- how were these 100 picked from the 111? And line 237

Line 251- you didn’t test this. It needs to be in the methods if you did AMR testing. If its only based on genes, then suggest that it had the genes to encode resistance but that this wasn’t checked

Line 272- this doesn’t make sense- perhaps- This study investigated the presence of pathogenic ….’

Comments on the Quality of English Language

These are detailed above

Reviewer 2 Report

Comments and Suggestions for Authors

Manuscript ID: animals-2801715

Dear Authors,

The topic addressed in this paper is interesting. However, in my opinion, the paper has some deficiencies and some sections need to be revised and integrated.

Below, I report my remarks:

INTRODUCTION

Line 66 and Line 70: In the scientific name also “spp” should be written in italics.

MATERIALS AND METHODS

Bacterial isolation and characterization

Lines 125-134: Although the ISO reference methods have been written in the materials and methods section, I suggest that the method used (e.g. the culture media used for isolation) be included in the text.

Lines 141-142: Table No. 3 is to be included and described in the "results" section.

Hepatitis E virus detection

Line 161-165: Insert details of the method used to detect the virus.

RESULTS

Sampling

Lines 214-218: The complete “sampling” section should be moved to the materials and methods section “sampling” as well as Table 5. In addition, with reference to lines 214 to 218, some of the data presented have already been inserted in the materials and methods section and should therefore not be repeated. Furthermore, with reference to table n.5, please specify in the text the criterion for choosing the sampled matrices.

Bacterial isolation and Characterization

Lines 223-228: The data presented are not clearly presented. Rephrase the text to make the data clearer and more understandable.

Line 233: After showing the number of samples on which the search for the germ was carried out, no mention is made of any positivity found, and the results relating to pathogenicity genes are mentioned directly. Rephrase and clarify this in the text.

Line 244-245: In Table n. 6 insert subtotals for liver and muscle to make the table clearer. Also in the column concerning positivity it is not clear which positivity is being referred to. Specify this in the text.

DISCUSSION

Lines 291-296: What does the presence of these virulence factors mean? What implications can they have for germs, animals and humans? Discuss these aspects in the text.

Lines 296-301: The presence of resistance to some antibiotics is mentioned but this is not discussed. Introduce reflections and discussions in the text.

Reviewer 3 Report

Comments and Suggestions for Authors

Wild ungulates can be carriers of STEC [14]. The comment about ungulates carrying pathogenic E. colis could be expanded further, it seems like a filler idea without context

Wild ruminants are a reservoir for 88 HEV, especially deer [17]. Likewise, from this brief text, it is introduced that we could comment on which species of ruminants have been detected or seen that may be carriers.

In the methodology section, the sequences of the primers used for gene detection or the reference from where they were taken are missing.

In the discussion section something could be added about which part of the tract or systems the pathogens that were searched for are found in greatest abundance.  

Round 2

Reviewer 1 Report

Comments and Suggestions for Authors

I wish to thank the authors for addressing my comments and wish them the best of luck for their future research 

Comments on the Quality of English Language

I have no further comments

Reviewer 2 Report

Comments and Suggestions for Authors

Manuscript ID: animals-2801715

Dear Authors,

The changes made to the manuscript are in accordance with what was requested. The manuscript is now clearer and more detailed.
